# Ovariectomy in Mouflons (*Ovis aries*) in the Field: Application of Innovative Surgical and Anaesthesiological Techniques

**DOI:** 10.3390/ani13030491

**Published:** 2023-01-31

**Authors:** Vincenzo Cicirelli, Alice Carbonari, Matteo Burgio, Francesca Giannini, Annalisa Rizzo

**Affiliations:** 1Department of Veterinary Medicine, University of Bari Aldo Moro, 70121 Bari, Italy; 2Natural Park of the Tuscan Archipelago, 57037 Portoferraio, Italy

**Keywords:** female mouflons, ovariectomy, midline approach, flank approach, CAIMAN^®^, field anesthesia

## Abstract

**Simple Summary:**

Often the overpopulation of wild animals can cause the destruction of the environment, for which the solutions are catches, culling, or birth control. Ovariectomy is a useful intervention for birth control. This report describes an innovative technique of gonadectomy, using a suitable anesthetic/analgesic protocol and an innovative device for the surgical ovariectomy, in the field conditions. This technique is simple to use, free of side effects, and can be used in a wildlife clinic.

**Abstract:**

This report describes an innovative technique of ovariectomy useful for the birth control in the mouflon population. Thirteen female mouflons in reproductive age were submitted to ovariectomy via midline and left flank, using the AESCULAP CAIMAN^®^ Seal and Cut device. The CAIMAN^®^ was useful for clamping the ovary, stapling the vessels, and cutting in one stroke, thus reducing the surgery times. The day after the gonadectomy all animals were mobilized to another enclosure. In this study, no intraoperative and post-operative complications were observed, and all animals were gonadectomized without side effects. This study sets the guidelines for the surgical sterilization of mouflons in the field using anesthetic protocols and high-quality surgical procedures.

## 1. Introduction

The mouflon (*Ovis aries*) is an artiodactyl mammal that belongs to the superorder Ungulates and the family Bovidae. It is a wild and highly adaptable species that descends from the Asian mouflon (*Ovis gmelini gmelinii*) and was introduced to Europe in the last century [1,2]. Since the Neolithic period in Italy, there has been an autochthonous population in Sardinia; in addition, the mouflon has been introduced and is widespread, with about 40 isolated populations (totaling about 5000 specimens) in some minor islands, including: Elba, Asinara, Capraia, Marettimo, Zannone, and Giglio [2]. In the reproduction field, mouflons are no different from sheep in the Northern Hemisphere. Being negatively affected by the photoperiod, the breeding season begins in October and ends in January, when the daily light hours increased. The gestation lasts for 155 days and deliveries occur between February and April, or between June and July giving birth to one to two lambs [3].

The attainment of puberty is dependent on the month of the birth; indeed, in wild animals, the spring-born lambs have been shown to reach puberty in the first available breeding season. In contrast, the animals born in late autumn must wait until the second available breeding season [4]. Mouflon rams are classified as adults at the age of 5 years, but endocrine and morphological changes continue throughout the life cycle. The difference in the attainment of puberty between the two sexes is present to promote the evolution of large physical features and conspicuous male characteristics, which allow greater reproductive success but which take several years to fully develop [5]. The introduction of mouflons in Europe has always been discouraged because these animals compete with other native species and may mate with domestic sheep [2]. In Italy, the mouflons present on Giglio Island create a variety of damages to the ecosystem and crops; moreover, their numbers are increasing more and more, making their management very complex [6]. Therefore, there is a need for a control plan with selective techniques designed not to harm non-target species, while at the same time reconciling the present gap between the population control and ethical issues. A plan for the eradication of the population has recently been approved by the relevant institutions, involving the use of selective harvesting techniques, bloodless methods (capture, sterilization, and translocation to fenced areas) [6]. After evaluating the different techniques used for birth control in other wild species (e.g., *Cervus elaphus*), such as tubal ligation and the administration of GnRH vaccines, the need for a definitive technique that would avoid having to capture the animals several times and induce the least stress led to the choice of the surgical ovariectomy [7]. The aim of this study was to describe the innovative surgical and anestesiological technique of the ovariectomy in female mouflons in the field condition, to control overcrowding in enclosures, and to avoid the selective culling of these animals. The two surgical approaches, flank and linea alba, were used, and the AESCULAP CAIMAN^®^ Seal and Cut device was employed for clamping the ovary, stapling the vessels, and cutting in one stroke. The hypothesis of this paper is that these practices are easy to perform, practical, and rapid so as to minimize the risks of interventions carried out in the field.

## 2. Materials and Methods

### 2.1. Ethic

This study was performed in accordance with the ethical guidelines of the Animal Welfare Committee. Institutional Review Board approval of the study was obtained from the University of Bari Aldo Moro, with approval number 20/2022.

### 2.2. Animals

This study enrolled 13 female mouflons. All of the translocated animals, with the exception of one animal born at the same Reserve, came from Giglio Island, as a result of activities conducted with the project LIFE18NAT/IT/000828 LETSGO GIGLIO “Less alien species in the Tuscan Archipelago: new actions to protect Giglio island habitats”. They were transferred to the Marsiliana Nature Reserve about a month before the surgery. The animals were housed in an enclosure of about three hectares, in which there was a funnel-shaped enclosure ending in a crate-shaped device with a guillotine opening to contain the animals (Figure 1). All animals were in good, alert condition and had a body condition score (BCS) of between 3 and 3.5 out of a maximum of 5 [8], and they were between 18 and 28 months old (mean ± standard deviation SD: 22.85 ± 3.55), and weighed between 15 and 18 kg (mean ± SD: 16.61 ± 1.19).

### 2.3. Anesthetic Protocol

Food and water were withheld from the patients for 24 h and 12 h prior to surgery, respectively. The day before the surgery, the animals were captured in a funnel-shaped enclosure to facilitate operations and minimize stress. The day of surgery, the animals were captured from the enclosure by three operators and contained for premedication. The anesthesia was performed using xylazine (Nerfasin^®^ 20 mg/mL, ATI, Ozzano dell’Emilia, Italy) 0.1 mg/kg and an association of tiletamine and zolazepam (Zoletil^®^ 50/50 mg/mL, Virbac, Milan, Italy) 4 mg/kg, mixed in the same syringe and injected in the brachiocephalicus muscle. After approximately 10 min, upon reaching a state of deep sedation, a 20-G venous catheter (DeltaVen^®^, DeltaMed S.p.A., Viadana, Italy) was inserted in the cephalic vein to start a maintenance fluid therapy (3 mL/kg/h of ringer with lactate, with possible variations during surgery, depending on the hemodynamic needs). Propofol (PropoVet Multidose^®^ 10 mg/mL, Zoetis Italia S.r.l., Rome, Italy) at 2 mg/kg was loaded into the syringe and was administered intravenously to allow orotracheal intubation, and anesthetic maintenance was performed with isoflurane (Isoflo^®^, Zoetis Italia S.r.l., Rome, Italy), vaporized in 100% oxygen, in an open anesthesia system (MedVet S.r.l., Taranto, Italy), always performed by the same anesthesiologist. All patients were connected to a re-breathing respiratory circuit and were allowed to breath spontaneously. This protocol was in accordance with that described by Caulkett and Haigh (2007) [9]. During the perioperative period, the animals were continuously monitored through multiparametric monitoring of the heart rate (HR), respiratory rate (RR), non-invasive blood pressure (BP), oxygen hemoglobin saturation (SpO_2_), and body temperature (T). In the event of increases in these parameters (>25% compared to the pre-incision values) during the procedure in response to the surgical pain, a bolus of fentanyl would be administered intravenously at 2 μg/kg (Fentadon^®^, Dechra Veterinary Products S.r.l., Torino, Italy) as rescue analgesia.

### 2.4. Surgical Procedures

All operations were performed by the same surgeon and by the same operating team in full compliance with the leges artis. Ten animals were gonadectomized using left flank as the surgical access. The paralumbar fossa from the transverse processes of the lumbar and sacral vertebrae and from the last rib to the level of the tuber coxae was clipped, shaved, and aseptically prepared. A vertical skin incision (about 6 cm long) with a number 23 scalpel blade was performed on the left flank on the paralumbar fossa close to the iliac wing (Figure 2). All muscular layers (in order, the external and internal abdominal oblique muscles and transverse muscle) were punctured with the scalpel to facilitate surgical access and then proceeded with the separation of the muscle fibers down to the peritoneum, which was held with the forceps, punctured with the scalpel, and cut with scissors. The surgeon grasped the uterus, using fingers, and exteriorized, locating the ovaries. The clamp of the CAIMAN^®^ (CAIMAN^®^ 5 non-articulated; Aesculap AG, Tuttlingen, Germany) vessel sealing device (handpiece 5 mm straight bite non-articulated jaw length 24 cm) was affixed to the base of the ovary, at the level of the ovarian pedicle, and then the ovariectomy was performed (Figure 3). Therefore, observation for 1 min of any hemorrhages from the remnant ovarian pedicle was carried out. All procedures were repeated on the other ovary. After the removal of the ovaries, the peritoneum and the abdominal transverse muscle were sutured together, and a second layer of sutures was used to close the internal and external abdominal oblique muscles. Both layers were sutured with synthetic absorbable suture USP2 (Surgicryl^®^ Polyglycolic Acid PGA, SMI, Belgium). The skin was closed with simple interrupted sutures using the same suture. In 3 female mouflons, the ovariectomy was carried out similarly, but using the linea alba access. The incision was performed forward at the udder, at about 10 cm from the umbilical scar (Figure 4). The skin was incised and the linea alba was identified. It was raised, by means of a surgical clamp, and an incision was performed with a scalpel. The incision was enlarged with scissors. The surgeon grasped the uterus, using fingers, and exteriorized, locating the ovaries. Thereafter, the procedures were similar to those described above. After the removal of the ovaries, the peritoneum and the linea alba were sutured together with a continuous suture using a synthetic absorbable suture thread USP2 (Surgicryl^®^ Polyglycolic Acid PGA, SMI, Belgium). The subcutaneous tissue and skin were closed with simple interrupted sutures using the same suture thread.

### 2.5. Pre-Operative and Post-Operative Procedures

Intramuscolar (IM) injection of 10,000 UI/kg benzilpenicilline and 12.5 mg/kg dihydrostreptomicine (Repen, Fatro S.p.A., Ozzano dell’Emilia, Italy; 200,000 UI + 250 mg/mL) was administered 10 min after the premedication to provide antibiotic coverage. Moreover, at the end of the surgery subcutaneous (SC) injection of triidrate amoxicilline (Betamox LA 150 mg/mL, Vètoquinol Italia S.r.l., Bertinoro, Italy) 15 mg/kg and the IM injection of 0.3 mg/kg ketoprofen (Zooketo 100 mg/mL, Elanco Italia S.p.A., Milan, Italy) were administered as the antibiotic and anti-inflammatory coverage for the post-operative period.

After the operations were completed, the animals were placed in special cases to ensure a peaceful and safe awakening. During this period, the animals were monitored by the veterinary staff.

Once fully awake, the animals were housed in a small stable within the enclosure.

All the animals were observed for 5 h post-surgery and the day after by the veterinary staff evaluating behavioral changes, such as reluctance to move, reduced feed intake, altered social interaction, and changes in posture. In case of the appearance of pain symptoms, IM injection of 0.3 mg/kg ketoprofen (Zooketo 100 mg/mL, Elanco Italia S.p.A., Milan, Italy) would be administered.

After that, the animals were released in a large enclosure and monitored by park operators for one week until the transfer to another reserve.

## 3. Results

Thirteen female mouflons were neutered. The flank approach ovariectomies were performed in 20.6 min (+6.65), while midline approach ovariectomies lasted 18 min (+1.73), from the start of the first skin incision to the placement of the last skin suture. The anesthesia of the female mouflons lasted, from the time of the premedication to the time of stopping of the isoflurane administration, an average of 46.54 min (±13.59). The animals were extubated approximately 7 min after the interruption of the isoflurane administration (6.54 ± 1.71) and recovered quadrupedal station approximately 15 min after extubation (14.92 ± 2.40). For all patients, no intra or post-operative complications were reported.

## 4. Discussion

This study aimed to describe a technique to gonadectomy useful for the birth control in the mouflon population. Our specific goal is to use the CAIMAN^®^ device that is normally used in the surgical room for the neutering of moufflons. The results of this study clearly show that field sterilization using the CAIMAN^®^ is possible with both the abdominal and flank surgical access. The duration of the procedure was about 25 min for both the procedures. This demonstrates that the surgical approach does not influence the duration of the procedure. No complications or side effects intra or post-surgery were observed in both groups. No relevant hemodynamic problems were observed during the procedures and no hemorrhages were observed during the ovariectomy in both groups, and ovaries were removed without any complications. Therefore, the seal created by the CAIMAN^®^ device is effective, sealing an intrinsic part of the tissue and vessel wall, which cannot be dislodged [10]. The use of a linear cutter improved the known procedure, as already described in other species [11]. Considering the relative simplicity of its execution, the routine use of this device is considered a useful alternative for wild animals’ neutering. Regarding surgery, the flank approach is an excellent technique, preferred by many authors, compared with the linea alba access. In our experience, the intraoperative/post-operative problems did not occur in any animals. For this reason, our opinion is that both surgical approaches used in our study are useful for a good surgical technique. However, it is important to underline that surgical access from the flank reduces the possibility of postoperative side effects, such as abdominal hernias caused by the volume of the rumen. Furthermore, this surgical access allows a better evaluation of the surgical wound at a distance [12].

Good anesthesia and appropriate pain management in veterinary patients is a crucial component of the appropriate patient care [9]. This has several advantages in wild animals, such as controlled anesthesia and extremely fast recovery. Furthermore, the intubation of the animals allows a reduction in the possibility of “polmonite ab ingests” due to any intraoperative regurgitation. Therefore, our study demonstrates that, even in the field, it is possible to have a suitable control of the quality of analgesia, using the multiparametric monitor. Anesthesia monitoring is useful for the determination of the specific moments the animals needed rescue analgesia, while preventing unnecessary ‘blind’ administrations. Furthermore, the rapid post-operative awakening has proved to be useful, for an optimal recovery of the vital activities of the animals. This aspect is to be underlined in the moufflons which are ruminant wild animals to avoid the occurrence of complications, such as ruminal tympanism and passive regurgitation [13]. In our study, no animal required intra or post-operative rescue analgesia: this demonstrates a suitable anesthetic protocol that aims to respect the animal welfare.

## 5. Conclusions

This is the first report on mouflon gonadectomy in the field. In this study, we used a suitable anesthesiologic and analgesic protocol, a constant pain assessment, and an optimal surgery technique. The anesthetic and surgical techniques described in this study are considered desirable for gonadectomy in moufflons because there were not any intraoperative complications or post-operative pain/discomfort. These results may provide the foundations for further investigations about the gonadectomy in wild animals, such as moufflons, which is useful for several purposes, such as the birth control.

## Figures and Tables

**Figure 1 animals-13-00491-f001:**
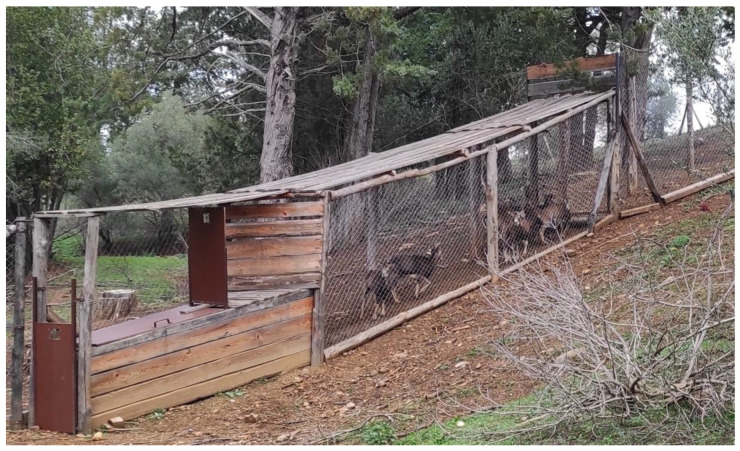
Funnel-shaped enclosure ending in a crate-shaped device with a guillotine opening to contain the animals.

**Figure 2 animals-13-00491-f002:**
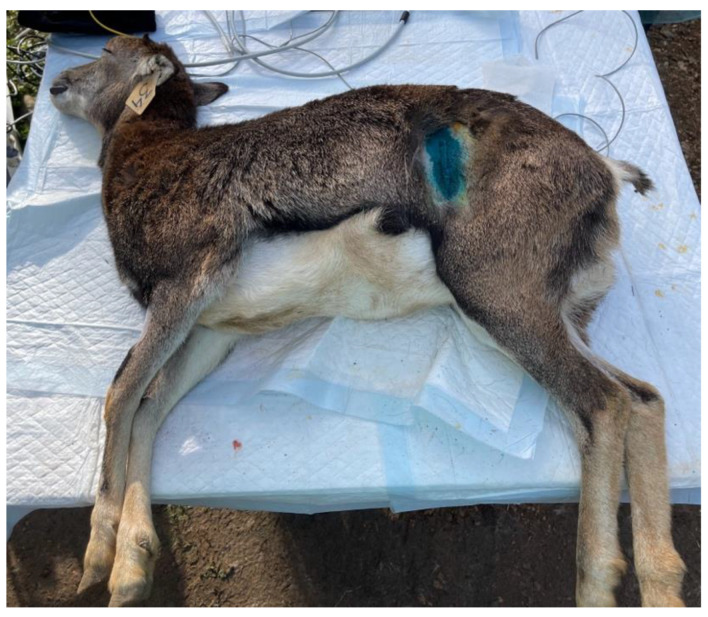
Area where the incision for left flank ovariectomy was made.

**Figure 3 animals-13-00491-f003:**
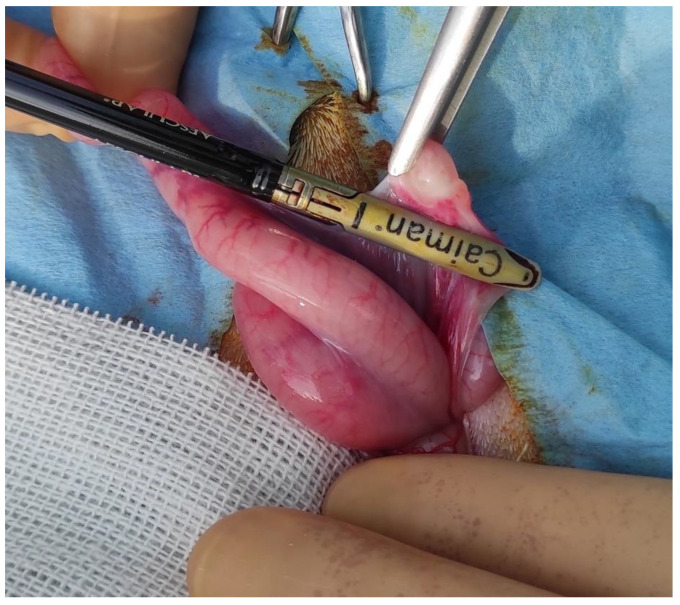
Individualization and exeresis of the ovary using CAIMAN^®^.

**Figure 4 animals-13-00491-f004:**
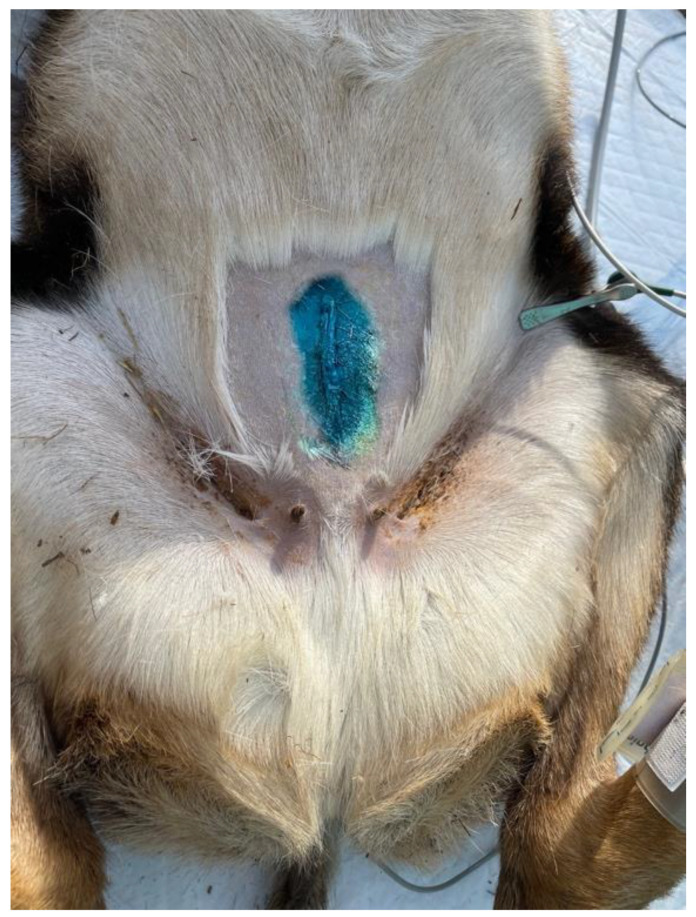
Area where the incision for linea alba ovariectomy was made.

## Data Availability

The data presented in this study are available on request from the corresponding author.

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
