# Peer review of "Ovariectomy in Mouflons (Ovis aries) in the Field: Application of Innovative Surgical and Anaesthesiological Techniques"

_animals, 2023, doi:10.3390/ani13030491_

Round 1

Reviewer 1 Report

Dear authors, I read your paper with great pleasure. 

The study in my opinion is well structured, the photos are clear and explanatory. 

The English is comprehensible although I have not made any judgements as I am not a native speaker.

In my opinion, the description of the surgical procedure is very precise, less so the anaesthesiological one.

More than once in the text, an excellent/optimal protocol is mentioned but in my opinion these phrases lead the reader to expect a very detailed description of the procedure. I believe that you should consider whether you should then either emplement the description of the anaesthesiological procedure or rephrase the sentences.

Simple summary

Linee 11: “…using an optimal anesthetic/ analgesic protocol…” is a strong term to use. Performing optimal anaesthesia means having to ensure a very detailed description of the procedure throughout the text. Consider reformulating or simply changing terms

Material and methods

Linee 80: “…months old (22,85±3,55) and weighed between 15 and 18 kg (16,61±1,19).”

Numbers presented in brackets represent mean and standard deviation? If so, I would suggest expressing it in the text;

Linee 93: “…standard maintenance fluid therapy (3 ml/kg/h of ringer with lactate)”

the use of the term “standard” in my opinion is forced. On the one hand, fluid therapy is a very dynamic therapy in the intraoperative phase and can often vary according to the anaesthetist's clinical assessment. I would be in complete agreement if you added a supporting reference. Otherwise I would say avoid the standard term and add a statement that fluid therapy could vary;

Linee 94: “….at 2 mg/kg was administered intravenously…”

All animals received the same dosage of propofol? Assuming all animals received 2 mg/kg, it is reasonable think that premedication protocol for every animals produce a state of sedation deep enough, it is correct?

have you thought about and considered a sedation score? In any case, considering the characteristics of propofol, unless exactly 2 mg/kg of propofol was administered to all 13 subjects, I would add the wording "2 mg/kg to effect until corrected state of intubation “or similar.

Linee 97: please specify system type and brand;

Discussion

Linee 188: “…our study demonstrates that even in the field it is possible to have excellent control of the quality of analgesia, using the multiparametric monitor.”

In order to say that you have managed correctly intraoperative nociception and post-operative pain, you should specify in the materials and methods section in great detail, the range of variation taken into account, the vital parameters and how many at the same time must exceed this range of variation. Did you take into account a 20 or 30% rise in at least 2 out of 3 vital parameters that you checked with the monitor? What kind of rescue analgesia did you plan to perform in case of a positive response to the nociceptive stimulus? It should be mentioned in the text even if none of your animals actually needed it. As for post-operative pain, how did you assess it? Did you apply a validated scale or did you create your own? If so, you should mention it and add it in the text. Again, what type of rescue analgesia did you choose to use in the case of a positive response to the scale?

Linee 195: “In our study no animal required intra o postoperative rescue analgesia: this demonstrates an excellent anesthetic protocol that aims to respect animal welfare.”

Please rephrase the sentence. “Excellent anesthetic protocol” does not seem to me an appropriate term for the meaning of the sentence.  Intraoperative nociceptive and post-op pain management is not the only goal of proper anaesthesia. For example, intraoperative pressure management. Do not mention this in any way, you probably did not experience any kind of management difficulties as hypotensive phases. However, more than once in the text you mention excellent protocol, safe protocol etc... so planning the anaesthesia protocol in all its steps is necessary. Have you considered the hypotensive cut-off? If yes, how would you have handled it? The same could be said for bradycardia, hypothermia and all those complications that can occur in such a procedure

Conclusion

Linee 200: “…Excellent protocol…” same considerations as above

Linee 202: “…that can be performed reducing the intraoperative risk”. Reduce the risk of complications or mortality? if so, reduce compared to what? is there a previous publication that considers higher complication and mortality rates than yours? if so, it should be cited and discussed

Best Regards

Author Response

Dear authors, I read your paper with great pleasure. 

The study in my opinion is well structured, the photos are clear and explanatory. 

The English is comprehensible although I have not made any judgements as I am not a native speaker.

In my opinion, the description of the surgical procedure is very precise, less so the anaesthesiological one.

More than once in the text, an excellent/optimal protocol is mentioned but in my opinion these phrases lead the reader to expect a very detailed description of the procedure. I believe that you should consider whether you should then either emplement the description of the anaesthesiological procedure or rephrase the sentences.

Thank You for your comments and suggestions, that we take in account, as You can see following.

Simple summary

Linee 11: “…using an optimal anesthetic/ analgesic protocol…” is a strong term to use. Performing optimal anaesthesia means having to ensure a very detailed description of the procedure throughout the text. Consider reformulating or simply changing terms

We changed the term.

Material and methods

Linee 80: “…months old (22,85±3,55) and weighed between 15 and 18 kg (16,61±1,19).”

Numbers presented in brackets represent mean and standard deviation? If so, I would suggest expressing it in the text;

We did it

Linee 93: “…standard maintenance fluid therapy (3 ml/kg/h of ringer with lactate)”

the use of the term “standard” in my opinion is forced. On the one hand, fluid therapy is a very dynamic therapy in the intraoperative phase and can often vary according to the anaesthetist's clinical assessment. I would be in complete agreement if you added a supporting reference. Otherwise I would say avoid the standard term and add a statement that fluid therapy could vary;

You are right. We eliminated the term "standard” and added a statement about fluid therapy.

Linee 94: “….at 2 mg/kg was administered intravenously…”

All animals received the same dosage of propofol? Assuming all animals received 2 mg/kg, it is reasonable think that premedication protocol for every animals produce a state of sedation deep enough, it is correct?

have you thought about and considered a sedation score? In any case, considering the characteristics of propofol, unless exactly 2 mg/kg of propofol was administered to all 13 subjects, I would add the wording "2 mg/kg to effect until corrected state of intubation “or similar.

After the administration of xylazine and tiletamine+zolazepam, we always waited until a state of deep sedation was reached before carrying out any manipulation of the animals since, being wild animals, they would become too stressed if they were not deeply sedated.

As for propofol 2mg/kg is the dosage loaded in the syringe for each animal; the amount administered, however, was to effect.

We have corrected the sentence in the text.

Linee 97: please specify system type and brand;

We did it

Discussion

Linee 188: “…our study demonstrates that even in the field it is possible to have excellent control of the quality of analgesia, using the multiparametric monitor.”

In order to say that you have managed correctly intraoperative nociception and post-operative pain, you should specify in the materials and methods section in great detail, the range of variation taken into account, the vital parameters and how many at the same time must exceed this range of variation. Did you take into account a 20 or 30% rise in at least 2 out of 3 vital parameters that you checked with the monitor? What kind of rescue analgesia did you plan to perform in case of a positive response to the nociceptive stimulus? It should be mentioned in the text even if none of your animals actually needed it. As for post-operative pain, how did you assess it? Did you apply a validated scale or did you create your own? If so, you should mention it and add it in the text. Again, what type of rescue analgesia did you choose to use in the case of a positive response to the scale?

We added more information about management of intraoperative nociception and post-operative pain.

As post-operative pain, in our knowledge, there are no any articles on this matter in mouflons, so we used generic pain evaluation, such as reluctance to move, reduced feed intake, altered social interaction and change in posture, in wild animals. 

Linee 195: “In our study no animal required intra o postoperative rescue analgesia: this demonstrates an excellent anesthetic protocol that aims to respect animal welfare.”

Please rephrase the sentence. “Excellent anesthetic protocol” does not seem to me an appropriate term for the meaning of the sentence.  Intraoperative nociceptive and post-op pain management is not the only goal of proper anaesthesia. For example, intraoperative pressure management. Do not mention this in any way, you probably did not experience any kind of management difficulties as hypotensive phases. However, more than once in the text you mention excellent protocol, safe protocol etc... so planning the anaesthesia protocol in all its steps is necessary. Have you considered the hypotensive cut-off? If yes, how would you have handled it? The same could be said for bradycardia, hypothermia and all those complications that can occur in such a procedure

We used the adjective “excellent” because we had not the occurrence of intra-and/or post-operative complications as a result of using this anaesthetic protocol in the field. However, we understand your suggestion, so we have removed the adjective. As regard the intraoperative management, in case complications arose, we would have acted professionally to all relevant needs, with emergency drugs and fluids.  

Conclusion

Linee 200: “…Excellent protocol…” same considerations as above

We deleted the adjective.

Linee 202: “…that can be performed reducing the intraoperative risk”. Reduce the risk of complications or mortality? if so, reduce compared to what? is there a previous publication that considers higher complication and mortality rates than yours? if so, it should be cited and discussed

You are right, but there are any similar studies in literature. So we rephrased the sentence.

Reviewer 2 Report

This is a good article that has value for the profession.  It should be published with a few modifications.

Surgical procedures:  

1. Document the names of the muscles that were encountered in the flank approach rather than simply saying "all muscular layers"  (line 109).

2.Clarify why the muscular layers were incised as opposed to simply separating muscle fibers on each layer. (line 109) Cutting the muscles may have a slightly greater risk of hernia of abdominal contents than an approach in which the muscle fibers are separated, as cutting the muscles reduces the beneficial effects of the fibers of internal and external abdominal oblique muscles being perpendicular to each other.

3.  Why suture the peritoneum? In dogs it has been shown that suturing the peritoneum increases the risk of the formation of adhesions.  Does the same thing occur in mouflons?

4. Line 124, "with forbs"?  Do you mean to say "forceps"?

5. Figure 3.  A larger picture with more detailed would be beneficial.

6. Observation period.  "All animals were observed for 5 hours post-surgery and the day after…"  Is this sufficient post-operative time to justify claiming that no complications or side effect(s) of intra or post-surgery were observed.

Results

1. Lines 159-160 state "For all patients no intra and post operative complications were reported."  They may very well be a true statement, but is it misleading. The observation period was confined to the day of and the day after surgery.  Was that observation period long enough to confidently state that there were no complications?

2.  The word "patience" (Line 159) should be patients.

Discussion

1. The point is made in the discussion that "surgical access from the flank eliminates the possibility of post-operative side effects, such as hernias…"  (Line 180).  Eliminate is too strong a word, it may reduce the incidence of side effects but not eliminate them.

2.  Line 196 "intra o postoperative" should be "intra or postoperative>"

3.  Be consistent Line 159 says "post operative," line 196 says "postoperative."  Which is it?

Author Response

This is a good article that has value for the profession.  It should be published with a few modifications.

Thank You for your comments and suggestions.

Surgical procedures:  

Document the names of the muscles that were encountered in the flank approach rather than simply saying "all muscular layers"  (line 109).

We have described all muscle that were encountered.

Clarify why the muscular layers were incised as opposed to simply separating muscle fibers on each layer. (line 109) Cutting the muscles may have a slightly greater risk of hernia of abdominal contents than an approach in which the muscle fibers are separated, as cutting the muscles reduces the beneficial effects of the fibers of internal and external abdominal oblique muscles being perpendicular to each other.

We did in this way. So we rephrased the sentence.

Why suture the peritoneum? In dogs it has been shown that suturing the peritoneum increases the risk of the formation of adhesions.  Does the same thing occur in mouflons?

Regarding suturing of the peritoneum in the literature there are conflicting opinions in the various animal species. Not having specific references for the mouflon, we preferred to suture it.

Line 124, "with forbs"?  Do you mean to say "forceps"?

Yes, we mean to say scissors. We have replaced the word.

Figure 3.  A larger picture with more detailed would be beneficial.

We changed it

Observation period.  "All animals were observed for 5 hours post-surgery and the day after…"  Is this sufficient post-operative time to justify claiming that no complications or side effect(s) of intra or post-surgery were observed.

You are right, but they didn’t stay in the small stable for a lot, being wild animals. So, the day after surgery, they were released in a large enclosure and monitored by park operators, for one week. This sentence was added in the text.

Results

Lines 159-160 state "For all patients no intra and post operative complications were reported."  They may very well be a true statement, but is it misleading. The observation period was confined to the day of and the day after surgery.  Was that observation period long enough to confidently state that there were no complications?

As write before, we added a sentence for explain this procedure.

The word "patience" (Line 159) should be patients.

We changed it

Discussion

The point is made in the discussion that "surgical access from the flank eliminates the possibility of post-operative side effects, such as hernias…"  (Line 180).  Eliminate is too strong a word, it may reduce the incidence of side effects but not eliminate them.

Correct observation. We have replaced "eliminates" with "reduces".

Line 196 "intra o postoperative" should be "intra or postoperative>"

We did it

Be consistent Line 159 says "post operative," line 196 says "postoperative."  Which is it?

We did it